# Effect of Acute High-Intensity Interval Training on Immune Function and Oxidative Stress in Canoe/Kayak Athletes

**DOI:** 10.3390/biology12081144

**Published:** 2023-08-18

**Authors:** Ting-Ting Lee, Tzai-Li Li, Bo-Jen Ko, Li-Hui Chien

**Affiliations:** 1Department of Aquatic Sports, University of Taipei, Taipei City 111036, Taiwan; tingtingleeee@gmail.com; 2Department of Sport Promotion, National Taiwan Sport University, Taoyuan 33301, Taiwan; leej@ntsu.edu.tw; 3Department of Physical Education, National Taichung University of Education, Taichung City 403514, Taiwan; seabook8080@gmail.com; 4Graduate Institute of Athletics and Coaching Science, National Taiwan Sport University, Taoyuan 33301, Taiwan

**Keywords:** antioxidant, immunoglobulin A, infectious disease

## Abstract

**Simple Summary:**

High-intensity interval training (HIIT) is a primary training method for canoe/kayak athletes to enhance physical performance. However, intense training regimens can cause immunosuppression due to inflammation, metabolic stress, and oxidative stress. This study aimed to examine the effects of kayaking/canoeing HIIT exercise on immune and oxidative stress measures in athletes. Results showed that acute sprinting interval training on a kayak can affect immune cell count and oxidative stress biomarkers. Coaches and sports science professionals should consider these findings, as high training frequency or inadequate recovery periods between HIIT sessions can temporarily suppress immune function.

**Abstract:**

The aim of this study was to investigate the effects of acute high-intensity interval training (HIIT) on immune function and oxidative stress in male canoe/kayak athletes who were well trained. A total of 22 participants were voluntarily recruited with an age range of 15.9 ± 2.3 years, height of 172.2 ± 5.5 cm, body mass of 63.30 ± 6.95 kg, and body fat of 13.77 ± 3.76%. The modified Wingate kayaking test on a kayak ergometer was performed by all participants. Blood samples were collected at three different time points: before the test (Pre-T), immediately after (Post-T), and 3 h post-test (Post-3 h). Saliva samples were collected at two different time points: before the test (Pre-T) and 3 h after the test (Post-3 h). Results indicated that acute canoe/kayak ergometry HIIT had significant effects on the percentages and counts of leukocytes, neutrophils, lymphocytes, and lymphocyte subsets. Additionally, it resulted in increased total LPS-stimulated neutrophil elastase release and alterations in plasma concentrations of superoxide dismutase, catalase, and TBARS. These findings suggest that conventional kayak HIIT regimens can have short-term effects on immune function and induce oxidative stress in athletes.

## 1. Introduction

Sprint canoe/kayak racing has been an Olympic sport since 1936, featuring events for men in distances of 200 m and 1000 m. Beginning with the 2024 Olympics, a 500 m event has also been added [1]. The energetic demands of these events have been calculated by high-level male kayakers during laboratory ergometry, and the results showed that the aerobic contributions were 37% and 82%, and the anaerobic contributions were 63% and 18% in the 200 m and 1000 m, respectively [2]. The race times appeared to range from 40 to 45 s in the 200 m [3] and from 3 to 4 min in the 1000 m events [4]. Flat-water sprint kayaking is a sport that mainly puts demands on the upper body and trunk musculature [5]. Previous studies have shown that flat-water kayakers require high values for maximal aerobic/anaerobic capacity and upper-body muscle strength [6,7]. Bishop [8] suggested that successful kayaking performance requires not only a high aerobic power but also anaerobic contribution. An elite kayak athlete usually has to compete in both distances rather than specializing in only one race [9]. Bishop [8] and Fernandez et al. [10] suggested that Olympic kayak athletes not only need a high aerobic capacity but also anaerobic power for successful performance. Therefore, high-intensity interval training (HIIT) is often one of the main training methods for canoe/kayak athletes to improve their physical performance.

HIIT is an aerobic exercise approach that usually involves alternating periods of high-intensity efforts with periods of rest or low-intensity exercise [11]. The rationale behind this strategy is to allow for the accumulation of a higher volume of vigorous exercise than would be possible with continuous high-intensity exercise [12]. Interval training (IT) has gained popularity in sports training over the past century. Coaches and athletes have widely adopted this training approach to train at workloads closer to their specific performance competition [13].

Moderate-intensity continuous training (MICT) of short to moderate duration (<60 min) is commonly acknowledged to improve immune defense [14]. Conversely, short-term episodes of high-intensity or high-volume aerobic exercise may induce temporary negative alterations in immune cell count and function, lasting from 3 to 72 h depending on the immune parameter [15,16]. This can result in immunosuppression, thereby increasing the risk of infectious diseases [14]. Evidence shows that elite endurance athletes are more susceptible to upper respiratory tract infections (URTI) after strenuous training and/or competition compared to sedentary counterparts [17]. Intensive training regimens that lead to increased levels of inflammation, metabolic stress, and oxidative stress typically result in immunosuppression [18].

Previous studies suggest that IT might induce changes in immune function for a few hours after exercise cessation [19,20,21]. There is evidence of both positive [22] and negative [23] functional adaptation of the immune system in response to HIIT, such as improvements in immune defense and reduced immune cell count or death. Acute aerobic exercise elicits changes in circulating leukocyte counts, mainly neutrophils and lymphocytes [24]. The neutrophil concentration in circulation increases during exercise and remains elevated for several hours post-exercise, but the degranulation and oxidative burst response to bacteria are reduced. Conversely, lymphocytes in circulation increase during exercise but may drop below the baseline during the recovery phase for 2–4 h [25]. During the recovery period after heavy exercise, lymphocyte subpopulations such as CD4+ T cells, CD8+ T cells, CD19+ B cells, CD16+ natural killer (NK) cells, and CD56+ NK cells may decrease, and the salivary secretion rate of immunoglobulin A (IgA) may also temporarily decrease [26]. Overtraining or training without sufficient recovery can further exacerbate these effects. The exercise-induced immunodepression may persist for a few hours or a few days after strenuous exercise or training [25,27].

On the other hand, strenuous exercise also increases the generation of free radicals, including reactive oxidative species (ROS) such as superoxide (O_2_·^−^), hydrogen peroxide (H_2_O_2_), hydroxyl radical (HO·), nitric oxide (NO), and peroxynitrite (ONOO-). Over the past two decades, it has been shown that heavy exercise increases the generation of these free radicals and the risk of oxidative damage to skeletal muscle [28]. Moreover, the stress induced by prolonged intense exercise may also substantially influence plasma concentrations of pro-inflammatory and anti-inflammatory cytokines [29]. However, antioxidants are molecules that prevent the cellular damage from oxidative stress including enzymatic antioxidants, such as superoxide dismutase (SOD), catalase, glutathione peroxidase (GSH-px), and lactoperoxidase (LP), and non-enzymatic, such as vitamin E, vitamin A, vitamin C, lactoferrin, and selenium. Previous studies have consistently shown that moderate exercise training elevates the levels of antioxidant enzymes and certain non-enzymatic antioxidants in muscle [30]. The impact of exercise on oxidative stress is a crucial area of research due to the increase in reactive oxygen species (ROS) that result from an exercise-induced rise in oxygen consumption. While a single bout of exercise can lead to oxidative stress, regular exercise training has been shown to decrease oxidative stress [31]. The extent to which exercise induces oxidative stress is dependent on various factors such as exercise type, mode, frequency, duration, and intensity [32]. Numerous studies have highlighted that acute aerobic exercise, especially when performed at high intensity levels, is associated with oxidative stress.

Canoe/kayak sport requires not only specific techniques but also a great deal of physical capacity. Therefore, canoe/kayak athletes often undergo intense interval training during the training period. However, it remains unclear whether the types of intensified training affect immune function and oxidative stress response. Therefore, the aim of this study was to investigate the evidence from clinical trials that investigated the acute effects of kayaking or canoeing HIIT exercise on immune and oxidative stress measures in canoe/kayak athletes. We conducted this investigation to determine the acute effects of HIIT on relevant immune parameters for which sufficient data were available.

## 2. Materials and Methods

### 2.1. Participants

Twenty-two well-trained male kayak athletes (age 15.9 ± 2.3 years, height 172.2 ± 5.5 cm, body mass 63.30 ± 6.95 kg, body fat 13.77 ± 3.76%) were recruited voluntarily as participants in this study. These athletes had undergone training for at least 3 years and had trained more than three times per week during the past three months. Participants were asked to attend a kayak training camp during the experimental period of this study, where they received the same training program from a coach. The training program consisted not only of kayaking but also of other activities such as running and strength training. The training volume ranged from 12 to 28 h per week. Athletes with allergies, immune diseases, and muscle injuries within the past six months were excluded. Before entering this study, subjects were asked to provide informed consent after the aims of this study and the experimental protocol were explained by the researchers. All participants were instructed to maintain their normal daily routine and to abstain from alcohol, smoking, medical treatment, and other nutritional supplements throughout the experiment. The study protocol was approved by the Medical Ethical Review Committee of Fu Jen Catholic University (FJU-IRB NO C102088).

### 2.2. Experimental Protocol

The participants were asked to arrive at the laboratory at 8 a.m. After their body composition was measured using bioelectric impedance metabolimetry (ioi353, Jawon Medical, Daejeon, Republic of Korea), they performed a modified Wingate kayaking test on a kayak ergometer (Dansprint ApS, Hvidovre, Denmark) following these procedures: after a 5 min standardized warm-up, the subjects were instructed to perform maximum intensity paddling for 90 s, repeated six times with a 4 min passive rest interval in between. Parameters were recorded every 90 s during the Wingate test. Blood samples were collected at four time points: before the test (Pre-T), immediately after the test (Post-T), and 3 h after the test (Post-3 h). Saliva samples were collected at three time points: before the test (Pre-T) and 3 h after the test (Post-3 h).

### 2.3. Blood Collection and Treatment

Blood samples were obtained by venipuncture from the antecubital vein and collected into vacutainer tubes. Specifically, 7 mL of blood was collected into a K3EDTA-treated tube and 7 mL into a heparin-treated tube. For the blood dispensed into the heparin vacutainer, 1 mL was immediately added into an Eppendorf tube to measure neutrophil degranulation. The blood collected into the K3EDTA vacutainer was used to measure blood composition and cell counts within 30 min. After centrifugation at 1500× *g* for 10 min at 4 °C, the remaining K3EDTA tubes were stored at −20 °C for later analysis of plasma. To obtain accurate measurements, the whole blood sample was stored at a temperature of 20–24 °C and lymphocyte subsets were measured within 6 h, with no centrifugation or agitation but gentle and multi-axis mixing before testing [33]. Total and differential leukocyte counts, as well as hematological analysis including hemoglobin and hematocrit, were measured using a hematology analyzer (Sysmex SE-9000, Sysmex Corporation, Kobe, Japan). Plasma volume changes were calculated from hemoglobin concentration and hematocrit measurements according to Dill and Costill [34].

### 2.4. Determination of Neutrophil Degranulation

The present study describes a methodology for analyzing elastase concentration in blood samples obtained from a lithium heparin vacutainer. Specifically, 1 mL of blood is immediately added to an Eppendorf tube (1.5 mL capacity) containing 50 µL of 10 mg/mL bacterial lipopolysaccharide (LPS) solution (Stimulant, Sigma, St. Louis, MO, USA). Following a gentle inversion, the mixture is incubated for 1 h at 37 °C with gentle mixing every 20 min. After the incubation period, the mixture is subjected to centrifugation for 2 min at 15,000× *g*. The supernatant is subsequently collected and immediately stored at –20 °C until further analysis. Elastase concentration is determined using enzyme-linked immunosorbent assay (ELISA) kits (PMN Elastase ELISA, IBL, Hamburg, Germany).

### 2.5. Determination of Lymphocyte Subsets

The whole blood samples were incubated at 18–20 °C for 15 min while being protected from light, after which erythrocytes were lysed using 2 mL of FACS lysing solution (BD, Franklin Lakes, NJ, USA). The resulting mixture was gently mixed and incubated for an additional 15 min, following which the tubes were centrifuged at 1500× *g* rpm for 5 min. The lysed red blood cells were then discarded, and the bands of lymphocytes were washed with 3 mL of PBS and sodium azide solution. After centrifugation at 1500× *g* rpm for 5 min at 18–20 °C, the supernatant was discarded, and the lymphocytes were suspended in 0.5 mL of PBS with 1% paraformaldehyde solution by gentle drawing. The samples were then analyzed by flow cytometry using a Cytomics FC500 Flow Cytometry machine (BD, Franklin Lakes, NJ, USA). To determine the lymphocyte subsets of CD4+, CD8+, CD19+, and CD56+, 100 μL of whole blood samples was mixed with a 25 μL antibody cocktail and separated into three 12 mm × 75 mm polypropylene falcon tubes. Each cocktail contained a combination of mouse anti-human antibodies for the following receptors: (1) 20 μL of FITC-conjugated CD4+ mixed with 5 μL of PerCP-Cy-conjugated CD8+ and (2) 20 μL of FITC-conjugated CD19+ mixed with 5 μL of PerCP-Cy-conjugated CD56+. For the isotype controls, (3) 20 μL of mouse anti-human FITC-conjugated IgG and 5 μL of PerCP-Cy IgG were used. All antibodies were purchased from BD Biosciences.

### 2.6. Index of Lipid Peroxidation

The level of lipid peroxidation in plasma samples was assessed by quantifying thiobarbituric acid reactive substances (TBARS). To prepare the TBARS reagent, 5 mL of 6 N NaOH was mixed with 18 g of trichloroacetic acid (TCA) and 456 mg of thiobarbituric acid (TBA) in 115 mL of distilled water (ddH_2_O). Subsequently, 100 μL of plasma was mixed with 1 mL of the TBARS reagent and incubated at 100 °C for 30 min. After incubation, the samples were immediately placed on ice for 10 min and then centrifuged for 10 min at 3900× *g* rpm. The supernatant was loaded into a 96-well plate, and the absorbance was measured spectrophotometrically at 535 nm.

### 2.7. Determination of Antioxidant Enzymes

The activity of superoxide dismutase (SOD) was determined using Sigma SOD Assay Kit, which employs a highly water-soluble tetrazolium salt (WST-1-2-(4-iodophenyl)-3-(4-nitrophenyl)-5-(2,4-disulfophenyl)-2H-tetrazolium, monosodium salt) that yields a water-soluble formazan dye upon reduction by a superoxide anion. Working solutions for WST were prepared by centrifuging the enzyme solution tube for 5 s, mixing by pipetting, and diluting 15 μL of enzyme solution with 2.5 mL of dilution buffer. Diluted SOD with dilution buffer was used as the standard. Next, 20 μL of sample solution was added to each sample and blank 2 well, and 20 μL of ddH2O was added to each blank 1 and blank 3 well. Then, 200 μL of WST working solution was added to each well and mixed carefully. Subsequently, 20 μL of dilution buffer was added to each blank 2 and blank 3 well, and 20 μL of enzyme working solution was added to each sample and blank 1 well, followed by thorough mixing. The plate was then incubated at 37 °C, and the absorbance was read at 450 nm using a microplate reader. Catalase activity was analyzed using the Catalase-520 OxisResearchTM Kit. HRP/Chromogen, hydrogen peroxide, and catalase standard were prepared according to the manufacturer’s instructions. Next, 30 μL of diluted standards or samples was added to tubes followed by the addition of 500 μL of substrate (10 mM H_2_O_2_) to each tube. The tubes were incubated at room temperature for exactly 1 min. Stop reagent (500 μL) was added to each tube, capped, and mixed by inversion. Then, 20 μL of each reaction mixture was transferred to cuvettes or tubes. Finally, 2 mL of HRP/Chromogen reagent was added to each cuvette or test tube and mixed by inversion. The mixture was incubated for 10 min at room temperature, and the absorbance was read at 520 nm.

### 2.8. Saliva Collection and Treatment

Participants remained seated throughout the saliva collection procedure. Prior to collection, they were instructed to clear their mouth by swallowing, after which whole saliva was collected by allowing it to passively drip into a pre-weighed sterile tube over a period of 2 min. Participants were instructed to abstain from drinking liquids for at least 10 min prior to each collection. Saliva flow rate was determined by weighing the collected samples. The assumed density of saliva was 1.0 g/mL. All samples were stored at a temperature of −20 °C for subsequent analysis [35].

### 2.9. Determination of Salivary Immunoglobin A

The concentration of salivary immunoglobulin A (sIgA) (mg·L^−1^) was determined using a sandwich ELISA method based on the protocol described by Li and Gleeson [36]. In brief, flat-bottomed microtitration plates were coated with the primary antibody, rabbit anti-human IgA (I-8760, Sigma), at a dilution of 1 in 800 in carbonate buffer (pH 9.6), and left overnight at 4 °C. After washing with phosphate-buffered saline (PBS, pH 7.2), the plates were blocked with a protein solution containing 2 g·L^−1^ bovine serum albumin in PBS. The saliva samples were diluted 1 in 500 with deionized water and analyzed in triplicate. A range of standards (human colostrum IgA, I-2636) up to 600 μg·L^−1^ was included for calibration. All samples from a single subject were analyzed on a single plate, which also incorporated standards. The plates were incubated for 90 min at room temperature. After washing, peroxidase-conjugated goat anti-human IgA (A-4165, Sigma) was added, and the plates were incubated for an additional 90 min at room temperature. After a final washing step, the substrate, ABTS (Boehringer Mannheim, Lewes, UK), was added and the absorbance was measured at 405 nm after 30 min. The secretion rate (µg·min^−1^) of sIgA was calculated by multiplying the sIgA concentrations by the saliva flow rate. Saliva was collected passively by dribbling into a pre-weighed sterile tube for 2 min while the subjects were seated, and they were instructed to swallow to empty their mouth before collection. Subjects were not allowed to drink anything for at least 10 min prior to each sample. Saliva flow rate (mL/min) was determined by weighing, and samples were frozen at −20 °C for later analysis [35].

### 2.10. Statistical Analysis

Statistical analyses were conducted using SPSS (version 20.0; SPSS Inc., Chicago, IL, USA). The results are reported as means ± standard deviation (SD). To investigate the effects of acute spring interval training on immune response and oxidative stress, a repeated measures one-way analysis of variance (ANOVA) was employed. The level of statistical significance was set at *p* < 0.05.

## 3. Results

To verify the impact of HIIT on athletes’ exercise intensity, we measured the athletes’ immediate Rating of Perceived Exertion (RPE) after each interval exercise. This allowed us to evaluate the exercise load resulting from HIIT. The RPE scores for the 22 participants across six exercise sessions are presented in Figure 1. The results of leucocyte count and function are presented in Table 1, which displays the mean and standard deviation of various measures including changes in total leukocyte, lymphocyte percentage, lymphocyte counts, neutrophil percentage, neutrophil counts, and neutrophil degranulation. The data indicated a significant increase in total leukocyte at Post-T and Post-3 h compared to Pre-T (*p* < 0.05). The lymphocyte percentage was significantly elevated at Post-T and then dropped below the baseline at Post-3 h (*p* < 0.05). Lymphocyte counts were significantly increased at Post-T and recovered to resting levels after three hours (*p* < 0.05). Neutrophil percentage was slightly decreased at Post-T and then elevated at Post-3 h (*p* < 0.05), and neutrophil counts were increased at Post-T and Post-3 h following canoe/kayak ergometry sprinting interval training (*p* < 0.05). Furthermore, this study revealed a significant increase in total LPS-stimulated elastase release at Post-T and Post-3 h after canoe/kayak ergometry sprinting interval training (*p* < 0.05). However, no changes were observed in LPS-stimulated elastase release per neutrophil.

Table 2 presents the mean and standard deviation in changes in lymphocyte subsets. The data revealed that the percentage of CD4+ cells significantly decreased at Post-T and returned to baseline level at Post-3 h (*p* < 0.05). Additionally, CD4+ cell count was significantly elevated at Post-T and then recovered at Post-3 h (*p* < 0.05). While there was no significant alteration in the CD8+ percentage, CD8+ cell counts were increased at Post-T (*p* < 0.05). Moreover, the percentage of CD19+ cells significantly decreased at Post-T and returned to baseline level at Post-3 h (*p* < 0.05). CD19+ cell counts were elevated at Post-T and then recovered at Post-3 h (*p* < 0.05). Additionally, both the percentage and number of CD56+ cells significantly increased at Post-T (*p* < 0.05). After canoe/kayak ergometry sprinting interval training, there was a significant increase in the CD4/CD8 ratio at Post-T (*p* < 0.05).

According to the results presented in Table 3, there was no statistically significant alteration observed in the concentration of saliva IgA, secretion rate of sIgA, or saliva flow rate.

The impact of acute high-intensity interval training on oxidative stress is presented in Table 4. A significant increase in superoxide dismutase was observed at Post-3 h (*p* < 0.05). Catalase levels exhibited a significant elevation at Post-T followed by a decrease below baseline (*p* < 0.05). Furthermore, a significant increase in TBARS was observed at Post-T (*p* < 0.05).

## 4. Discussion

Our purpose was to examine the effects of acute HIIT on immune function and oxidative stress in canoe/kayak athletes. The present study revealed that acute canoe/kayak ergometry HIIT has significant effects on the percentages and counts of leukocytes, neutrophils, lymphocytes, and lymphocyte subsets. Moreover, it results in an increase in the total LPS-stimulated neutrophil elastase release and alterations in the plasma concentrations of superoxide dismutase, catalase, and TBARS.

The results of the present study indicate that acute sprinting interval training on a kayak alters the circulating numbers of immune cells. The total number of leukocytes increased immediately post-exercise and 3 h post-exercise. During the recovery period, the percentage of circulating lymphocytes declined below resting levels, while circulating numbers and the percentage of neutrophils continued to increase at least until 3 h post-exercise. The precise mechanisms of leukocytosis during and after exercise have not been fully elucidated; however, previous studies have suggested that elevated leukocytes may be flushed out from marginal pools such as the lungs, spleen, and vascular wall [37]. Some studies have further indicated that catecholamines and cortisol may act as mediators of this process [38,39]. Several studies have confirmed transitory increases in total leukocyte count lasting up to 6 h after HIIT [40,41]. The decline of total lymphocyte numbers may be induced by stress hormones, including epinephrine, norepinephrine, and cortisol.

Previous research suggests that catecholamines induce the initial increase in lymphocyte counts [42] and cortisol induces lymphopenia post-exercise [43]. In addition to the numbers of circulating leukocytes, the function of leukocytes is also an equally important consideration. The results of this study found that neutrophil counts were significantly increased at post-exercise and during the recovery phase. Numerous studies have reported an elevation in neutrophil count following HIIT performance [40,41,44]. This increase in neutrophil count typically occurs immediately after exercise and persists between 30 min and 5 h [40,41]. Other studies have also confirmed a delayed increase in this parameter, which occurs between 1 and 3 h after exercise [45,46].

Neutrophils represent about 60% of circulating leukocytes and are an important part of innate immunity. They migrate to infectious sites and bind, ingest, and kill pathogens through oxidative burst and degranulation [43]. Previous studies have indicated that acute exercise may elicit an increase in plasma elastase concentration following high-intensity exercise [47]. In the study by Souza et al. [48], it was found that while neutrophil count increases after an IT session, neutrophil function experiences a transient reduction upon in vitro stimulation [19,20]. This temporary functional impairment may be partially attributed to the heightened release of functionally immature neutrophils from the bone marrow or direct factors such as stress hormones and oxidative stress [19]. However, it remains unclear whether this brief functional impairment holds clinical significance. It is noteworthy that neutrophil function fully recovers 24 h after exercise [20]. Although stress hormones were not measured in this study, Rowbottom and Green [49] also suggested that the mechanism behind the increase in neutrophil counts during the short recovery period may be mediated by circulating stress hormones.

In this study, the LPS-stimulated elastase release per neutrophil throughout the experimental period did not change. This finding was not consistent with previous studies, which reported that a single bout of 2 h cycling at 60% VO_2max_ reduced the response of neutrophil degranulation to LPS stimulation in vitro on a per-cell basis [50]. This difference may be due to different exercise patterns, such as duration and intensity, and factors associated with neutrophil degranulation, such as the level of intracellular cAMP, phagocytic activity, platelet–neutrophil contacts, adrenaline, glucocorticoids, and IL-6 [36].

Lymphocyte subsets studied in this research included T helper cells (CD4+), T cytotoxic cells (CD8+), B cells (CD19+), and natural killer cells (CD56+). It has been suggested that some cell types may be more susceptible to hormonal influences than others. The findings of the present study demonstrated similar results to a previous study, which showed that NK cell counts increased to a greater extent than T cells and B cells, and CD8+ cells increased to a greater extent than CD4+ cells [51]. This study reported a typical pattern of cellular redistribution, which decreased the CD4+ percentage and increased CD56+ percentage correspondingly during and after exercise [52,53]. The percentage of CD19+ and CD8+ appeared to be less affected in this study. Notably, the CD4/CD8 ratio, which is thought to have clinical importance, was increased in this study. T cells are largely responsible for integrating cellular immunity. Many studies have reported significant reductions in T cell proliferation after repeated interval sprint exercises [54,55,56]. Frisina et al. [55] reported that T cell proliferation after exercise was negatively correlated with the percentage of NK cells. The alteration of the CD4+ percentage was larger than the CD8+ percentage, and this alteration may further affect the CD4+/CD8+ ratio. The B cell response can be a combination of cellular proliferation and differentiation, as well as the production of immunoglobulin capable of binding to specific antigens [54]. The results of the present study showed that the CD19+ percentage decreased during interval sprinting exercise. Immunoglobulin is synthesized by plasma cells, and salivary IgA is regarded as the first line of defense against many pathogens. Many studies in athletes have reported a decrease in salivary IgA after high-intensity exercise [57,58]. However, this study showed that salivary IgA concentration and secretion rate did not change, although the CD19+ percentage and counts were altered by an acute interval sprint exercise. The result of the present study found that CD56+ increased significantly post-exercise and returned to resting levels within 3 h after exercise, which was similar to the findings of previous studies [24,59]. NK cells are recognized as the first line of defense against many tumors and virus-infected cells, as they are able to initiate cytotoxic killing without prior sensitization. It has already been observed that acute exercise causes a significant increase in circulating counts of NK cells by as much as 400% [52,53]. Furthermore, since circulating NK cell counts increased to a greater extent than other lymphocyte subsets, the CD56+ percentage may increase from around 15 to 30% of lymphocytes [38]. In a review article by Souza et al. [48], it is suggested that the analysis of lymphocyte subsets indicates that HIIT may not necessarily result in lymphopenia. However, even in the absence of lymphopenia, lymphocytes may still become more susceptible to stressors in the few hours following an HIIT session [21,40,53]. These temporary functional impairments could be partly attributed to changes in lymphocyte subsets, such as a decrease in CD4+ and an increase in natural killer cells, leading to an impaired response to specific antigens [53,56]. The impaired function of lymphocytes during IT recovery may be due to direct mechanisms, such as lymphocyte redox imbalance [21] and stress hormone production [60].

There was no statistically significant alteration observed in the concentration of saliva IgA, secretion rate of sIgA, or saliva flow rate. Previous studies have reported varying results regarding the acute effect of exercise on salivary IgA, which could be attributed to a wide range of methodological aspects across the studies analyzed. These aspects include factors such as gender, sex hormones, and autonomic nervous system activity [61,62]. Furthermore, other methodological issues, such as dehydration, the method of saliva collection, and the expression of IgA, may also contribute to the discrepancies between studies, as previously noted [18]. Souza et al. [48] also emphasized that the acute impairments in the salivary secretory IgA rate that were observed after a single session of HIIT were not associated with an increased risk of upper respiratory tract infection (URTI) [63,64]. This finding suggests that the magnitude of the transient depression in salivary secretory IgA after HIIT does not have any clinical significance.

The results of this study show that TBARS concentrations increased post-exercise and then returned to baseline at 3 h post-exercise. TBARS is malondialdehyde (MDA) produced by lipid peroxidation and is commonly studied as a marker of lipid peroxidation during exercise. It has been consistently demonstrated that a single bout of various types of exercise induces lipid peroxidation, including anaerobic running [65] and cycling sprints [66], aerobic [67], and mixed aerobic–anaerobic exercises [68]. An acute bout of exercise is known to increase the activities of antioxidant enzymes, including superoxide dismutase (SOD), catalase, and GSH peroxidase (GPX), in skeletal muscle, heart, and liver [30]. The results of this study show that plasma SOD and catalase concentrations increased after a single bout of interval exercise. However, the data on antioxidant enzymes’ response to acute exercise are equivocal, with some studies showing an increase [69], others a decrease [70], and still others no change [71]. The inconsistent results may be due to the type of exercise, fitness levels of participants, and timing of blood samples [70,72]. However, the significant alteration of lipid oxidation and antioxidant levels suggests that acutely interval sprint exercise training significantly induces oxidative stress in canoe/kayak athletes (Figure 2).

The limitations of this study include its restricted collaboration with local coaches, resulting in a relatively small sample size. The 22 well-trained male kayak athletes recruited for this study were all from the same local county and underwent similar training. Their physical fitness may not fully represent all kayak athletes. Furthermore, the athletes’ age ranged from 13 to 21 years old, encompassing a significant range of physical growth and differences in athletic performance, which could potentially influence the results. However, a strength of this study is that there is limited research on HIIT specifically targeting adolescent kayak athletes. Future studies could also extend to observing the effects of HIIT on cardiovascular responses and overall physical fitness in adolescent kayak athletes.

## 5. Conclusions

This study aimed to investigate the effect of acute HIIT on immune function and oxidative stress in canoe/kayak athletes. The data obtained in this study demonstrated that the conventional kayak HIIT regimen can have short-term effects on the immune function of athletes and induce oxidative stress. These findings may have practical implications for coaches and sports science professionals to consider, as high training frequency or inadequate recovery periods between HIIT sessions may lead to a temporary suppression of immune function. Conversely, appropriately prescribing HIIT and providing sufficient rest periods may lead to improved physical performance while also preserving or enhancing the immune system. It is suggested that future research could explore the application of nutritional supplementation or alternative approaches to support athletes’ recovery. This research is anticipated to contribute to training practices.

## Figures and Tables

**Figure 1 biology-12-01144-f001:**
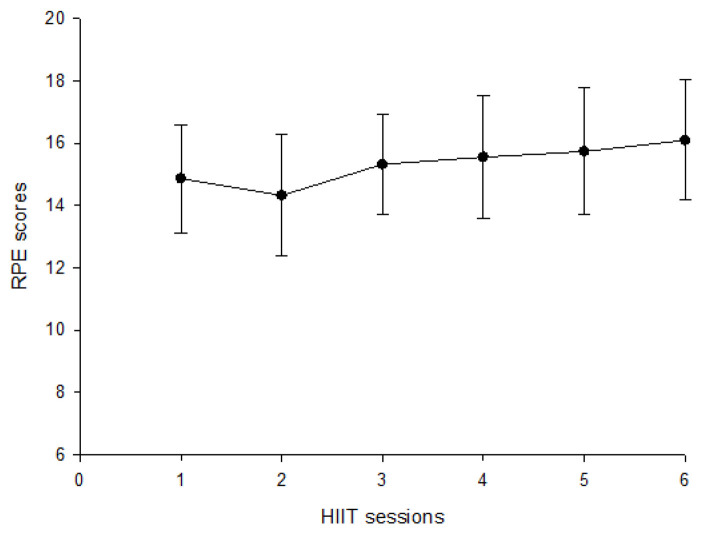
RPE scores across six exercise sessions in the HIIT.

**Figure 2 biology-12-01144-f002:**
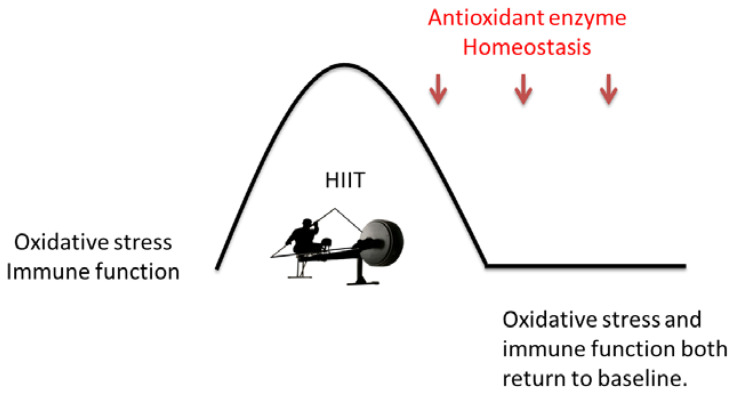
The potential mechanisms of changes in oxidative stress and immune function.

**Table 1 biology-12-01144-t001:** Change in leucocyte count and function.

Variables	Pre-T	Post-T	Post-3 h
Total leucocytes (10^9^·L^−1^)	5.09 ± 1.15	9.04 ± 2.12 *	8.47 ± 1.94 *
Lymphocyte percentage (%)	35.46 ± 8.11	45.73 ± 8.66 *	21.05 ± 7.18 ^#^
Lymphocyte counts (10^9^·L^−1^)	1.77 ± 0.44	4.14 ± 1.27 *	1.71 ± 0.43 ^#^
Neutrophil percentage (%)	53.53 ± 8.87	44.11 ± 9.21 *	72.41 ± 8.16 *^#^
Neutrophil counts (10^9^·L^−1^)	2.79 ± 0.96	4.02 ± 1.28 *	6.24 ± 2.04 *^#^
Total LPS-stimulated elastase release (ug·L^−1^)	232.72 ± 91.80	297.90 ± 108.86 *	486.30 ± 205.09 *^#^
LPS-stimulated elastase release per neutrophil (fg·cell^−1^)	84.94 ± 26.36	74.72 ± 14.18	79.27 ± 26.38

* Significantly from Pre-T (*p* < 0.05); ^#^ significantly from Post-T (*p* < 0.05).

**Table 2 biology-12-01144-t002:** Change in percentage and counts of lymphocyte subsets.

Variables	Pre-T	Post-T	Post-3 h
CD4^+^ percentage (%)	33.95 ± 10.37	23.23 ± 8.95 *	35.24 ± 11.58 ^#^
CD4^+^ counts (10^9^·L^−1^)	0.60 ± 0.20	0.95 ± 0.45 *	0.59 ± 0.22 ^#^
CD8^+^ percentage (%)	25.03 ± 10.92	22.85 ± 9.81	25.07 ± 10.21
CD8^+^ counts (10^9^·L^−1^)	0.41 ± 0.23	0.89 ± 0.41 *	0.38 ± 0.23 ^#^
CD19^+^ percentage (%)	14.16 ± 6.23	9.32 ± 4.26 *	15.56 ± 6.43 ^#^
CD19^+^ counts (10^9^·L^−1^)	0.26 ± 0.14	0.40 ± 0.18 *	0.27 ± 0.12 ^#^
CD56^+^ percentage (%)	11.24 ± 5.91	25.97 ± 14.22 *	9.40 ± 6.78 ^#^
CD56^+^ counts (10^9^·L^−1^)	0.19 ± 0.12	1.17 ± 0.91 *	0.16 ± 0.12 ^#^
CD4/CD8 ratio	1.63 ± 0.75	2.93 ± 1.85 *	1.70 ± 1.00 ^#^

* Significantly from Pre-T (*p* < 0.05); ^#^ significantly from Post-T (*p* < 0.05).

**Table 3 biology-12-01144-t003:** Change in secretion of saliva immunoglobulin A.

Variables	Pre-T	Post-T	Post-3 h
Saliva flow rate (mg·min^−1^)	263.89 ± 103.23	282.46 ± 114.79	241.12 ± 119.32
SIgA concentration (mg·L^−1^)	0.39 ± 0.40	0.26 ± 0.17	0.37 ± 0.25
SIgA secretion rate (mg·min^−1^)	0.07 ± 0.04	0.08 ± 0.07	0.08 ± 0.07

Abbreviation: SIgA = saliva immunoglobulin A.

**Table 4 biology-12-01144-t004:** Change in blood SOD, CAT, and TBARS.

Variables	Pre-T	Post-T	Post-3 h
SOD (U·mL^−1^)	3.61 ± 2.48	4.03 ± 3.22	4.84 ± 3.42 *^#^
CAT (U·mL^−1^)	506.16 ± 191.11	693.09 ± 202.62 *	337.54 ± 121.82 *^#^
TBARS (U·mL^−1^)	2.51 ± 0.72	3.08 ± 1.17 *	2.40 ± 0.55 ^#^

Abbreviations: SOD = superoxide dismutase; CAT = catalase; TBARS = thiobarbituric acid reactive substances. * Significantly from Pre-T (*p* < 0.05); ^#^ significantly from Post-T (*p* < 0.05).

## Data Availability

The data presented in this study are available on request from the corresponding author.

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
