# Peer review of "Effect of Acute High-Intensity Interval Training on Immune Function and Oxidative Stress in Canoe/Kayak Athletes"

_biology, 2023, doi:10.3390/biology12081144_

Round 1

Reviewer 1 Report

Manuscript by Lee et al., have investigated the effects of acute high-intensity interval training on immune function and oxidative stress in male canoe/kayak athletes. After analyzing the blood and saliva samples from the participants they found that acute canoe/kayak ergometry HIIT had significant effects on the percentages and counts of leukocytes, neutrophils, lymphocytes, and lymphocyte, increased total LPS-stimulated neutrophil elastase release and alterations in plasma concentrations of superoxide dismutase, catalase, and TBARS. Authors suggest that conventional kayak HIIT regimens can have short-term effects on immune function and induce oxidative stress in athletes. This is an important information for the athletes.

Quality of English is good, minor corrections are required.

Author Response

Thank you for the reviewer's feedback. We will further refine the English editing.

Reviewer 2 Report

The authors conducted a single-arm, quasi-experimental (pre-post) study to evaluate the effect of a kayaking HIIT intervention on biomarkers related to immune function and oxidative stress. The authors report finding an increase in the biomarker levels suggesting that the intervention is associated with an acute increase in the biomarkers related to immune function. Suggestions for improvement:

1) Was heart rate measured to ensure that the HIIT intervention caused its intended impact? This may be important because of potential for conditioning in experienced athletes.

2) Was multiple comparisons adjustment done for reporting multiple p-values?

3) Please consider presenting results as a bar graph.

4) Please consider adding a figure showing potential biologic mechanisms that explain these associations

5) Please discuss methodological limitations of this study- e.g. small sample size, selection bias, potential for bias from confounders.

6) In the discussion, please mention the implications or use of these findings- how will these findings inform practice or next steps for research?

N/A

Author Response

Thank you for the reviewer's feedback. Please see the attachment.

Reviewer 3 Report

I congratulate the authors for studing on such a specific group. But the content is very interesting, but I think there are some things to consider prior to publication. Firstly, sample size calculation is missing entirely from your methods, and this could question the power of your study. Most importantly, the way you have handled your data can also be questioned. 

While your article touches upon an important topic regarding the effects of acute HIIT on immune function and oxidative stress in canoe/kayak athletes, there are certain aspects that have contributed to my decision. The study has several limitations, including a lack of information on sample size and participant characteristics (age), which affects the generalizability of the findings.

Additionally, While your article addresses an important topic related to the effects of acute HIIT on immune function and oxidative stress in canoe/kayak athletes, I found that the novelty and originality of the study are limited. Several previous studies have already examined similar aspects of immune function and oxidative stress in response to exercise, including high-intensity interval training. Although your study provides additional insights, it lacks substantial new contributions to the existing body of knowledge in the field.

Author Response

Thank you for the reviewer's feedback. Indeed, as you mentioned, the limitations of this study include its restricted collaboration with local coaches, resulting in a relatively small sample size. The 22 well-trained male kayak athletes recruited for this study were all from the same local county and underwent similar training. Their physical fitness may not fully represent all kayak athletes. Furthermore, the athletes' age ranged from 13 to 21 years old, encompassing a significant range of physical growth and differences in athletic performance, which could potentially influence the results. However, the results of this study still demonstrate that regular HIIT in kayak athletes induces oxidative stress and potentially affects post-exercise immune function. It is suggested that future research could explore the application of nutritional supplementation or alternative approaches to support athletes' recovery. This research is anticipated to contribute to training practices.

Reviewer 4 Report

The purpose of the reviewed manuscript was to measure the effects of HIIT kayak/canoe exercise on measures of immune and oxidative stress in athletes. To achieve this goal, the authors used an experimental design that included HIIT with passive recovery, based on the Wingate test. The selection of the research group and method was appropriate to the research objective. The material and methods are described in detail and accurately. I just think that addressing the issue based on the Open Window theory is a mistake and a limiting factor. I suggest carefully reviewing all the latest Open Window evidence, even the most critical, and rewriting all related parts. Statistical analysis was successfully performed using appropriate tests. The results were presented graphically in a legible way. The authors correlated their research findings extensively with the available literature. They also provided the application of the results obtained towards a practical application towards the coaches.

Author Response

Thank you for the reviewer's feedback. Upon further review of previous studies, it is indeed evident that there are no precedents explaining the immunological changes due to HIIT through the open window theory. Additionally, the data from this study did not observe similar phenomena. We appreciate the reviewer's feedback and will proceed to remove this section.

Reviewer 5 Report

This is very interesting, well-written study. I rate it positive. However, some issues need to be addressed before publication.

Introduction

110 – Before the last paragraph I miss the summary of the introduction which would indicate justification of your study

119 – Please emphasize the potential practical application of Your study

Material and Methods

121/129 – I understand that data from 22 individuals were included in analysis. But this description need to be re-write. How many athletes were excluded? Clarify, please. Firstly, indicate how many subjects were involved as a potential participants, then how many were excluded due to inclusion-exclusion criteria, and then how many were included in the final analysis.

140 – Briefly describe the warm-up.

252 – Did you check data normality?

255 – Did you check variance homogeneity?

Results

136 – Did you measure the intensity of effort? Despite it was the instruction of “maximum intensity paddling” the information of objective (HR rate) or subjective (RPE scale) individual performance would add more value to this study.

257 – Tables are not positioned in places where they are referred for the first time. Therefore, results section is hard to follow.

Discussion

419 – Provide limitations and strength of your study. Please also indicate the future direction studies that could develop your observation. I suggest considering the individual response on physical effort intervention:

Domaradzki J, Koźlenia D, Popowczak M. Prevalence of Positive Effects on Body Fat Percentage, Cardiovascular Parameters, and Cardiorespiratory Fitness after 10-Week High-Intensity Interval Training in Adolescents. Biology (Basel). 2022;11(3):424. Published 2022 Mar 10. doi:10.3390/biology11030424

Maturana FM, Schellhorn P, Erz G, et al. Individual cardiovascular responsiveness to work-matched exercise within the moderate- and severe-intensity domains. Eur J Appl Physiol. 2021;121(7):2039-2059. doi:10.1007/s00421-021-04676-7

Conclusions

430 – Develop the practical application of your study. What information, your study, bring to practitioners? How your results can be utility?

Author Response

(The authors gave the same response as above.)

Round 2

Reviewer 3 Report

The revised version is acceptable for publication.